# Cold Bias of ERA5 Summertime Daily Maximum Land Surface Temperature over Iberian Peninsula

**Frederico Johannsen** [1]**, Sofia Ermida** [1,2]🆔**, João P. A. Martins** [1,2]🆔**, Isabel F. Trigo** [1,2]🆔**, Miguel Nogueira** [1]🆔 **and Emanuel Dutra** [1,*]🆔

[1] Instituto Dom Luiz, IDL, Faculty of Sciences, University of Lisbon, 1749-016 Lisbon, Portugal; fc46567@alunos.fc.ul.pt (F.J.); sofia.ermida@ipma.pt (S.E.); joao.p.martins@ipma.pt (J.P.A.M.); isabel.trigo@ipma.pt (I.F.T.); mdnogueira@fc.ul.pt (M.N.)

[2] Instituto Português do Mar e da Atmosfera (IPMA), 1749-77 Lisbon, Portugal

[*] Correspondence: endutra@fc.ul.pt

**Abstract:** Land surface temperature (LST) is a key variable in surface-atmosphere energy and water exchanges. The main goals of this study are to (i) evaluate the LST of the European Centre for Medium-Range Weather Forecasts (ECMWF) ERA-Interim and ERA5 reanalyses over Iberian Peninsula using the Satellite Application Facility on Land Surface Analysis (LSA-SAF) product and to (ii) understand the main drivers of the LST errors in the reanalysis. Simulations with the ECMWF land-surface model in offline mode (uncoupled) were carried out over the Iberian Peninsula and compared with the reanalysis data. Several sensitivity simulations were performed in a confined domain centered in Southern Portugal to investigate potential sources of the LST errors. The Copernicus Global Land Service (CGLS) fraction of green vegetation cover (FCover) and the European Space Agency's Climate Change Initiative (ESA-CCI) Land Cover dataset were explored. We found a general underestimation of daytime LST and slightly overestimation at night-time. The results indicate that there is still room for improvement in the simulation of LST in ECMWF products. Still, ERA5 presents an overall higher quality product in relation to ERA-Interim. Our analysis suggested a relation between the large daytime cold bias and vegetation cover differences between (ERA5 and CGLS FCocver) with a correlation of −0.45. The replacement of the low and high vegetation cover by those of ESA-CCI provided an overall reduction of the large Tmax biases during summer. The increased vertical resolution of the soil at the surface, has a positive impact, but much smaller when compared with the vegetation changes. The sensitivity of the vegetation density parameter, that currently depends on the vegetation type, provided further proof for a needed revision of the vegetation in the model, as there is a reasonable correlation between this parameter and the Tmax mean errors when using the ESA-CCI vegetation cover (while the same correlation cannot be reproduced with the original model vegetation). Our results support the hypothesis that vegetation cover is one of the main drivers of the LST summertime cold bias in ERA5 over Iberian Peninsula.

**Keywords:** land surface temperature; remote sensing; reanalysis; ECMWF

## 1. Introduction

Land Surface Temperature (LST) is a key variable for the surface-atmosphere energy and water exchanges and it was recently integrated as an Essential Climate Variable (ECV) into the Global Climate Observing System (GCOS) [1]. LST may be retrieved from remote sensing observations performed with channels sensitive to the radiance emitted by the land surface, i.e., usually within the thermal infrared (TIR) or in the microwave (MW) regions of the electromagnetic spectrum. Remotely sensed LST is defined as the radiometric temperature (due to its derivation from the radiance emitted by the

planet's surface), which is the temperature of the surface layer whose depth is equal to the penetration depth of the radiation used in its determination [2,3]. In the case of TIR radiation, the penetration depth is less than 1mm [4], which is why this variable is also referred to as "skin temperature".

The use of satellite LST has been steadily increasing during the last decades, from the evaluation and improvement of land surface models [5–8] to filling gaps in 2-meters air temperature (T2m), particularly in areas where station coverage is poor [9]. The latter has great potential to improve the quality of T2m observation datasets [10]. This is relevant due to LST's globally available datasets, while T2m is only measured in in-situ stations.

There are several methods to derive LST from remote sensing observations. Using TIR observations, the most common algorithm is the Generalized Split-Window [11,12], applied for example by the Satellite Application Facility on Land Surface Analysis (LSA-SAF) [13], in the estimation of the Moderate Resolution Imaging Spectroradiometer (MODIS) LST products [14] and in the Copernicus Global Land Service (CGLS) LST product [15]. One of the limitations of TIR-based LST data is its dependence on clear-sky measurements. Absence of LST data will occur for pixels classified as totally or partially cloudy during the observation period, and therefore such satellite LST products will be biased towards clear-sky conditions [16]. This implies that any evaluation of model LST using TIR-based products must be preceded by a careful cloud screening in the model dataset to ensure the compatibility of model and satellite variables.

Land surface is one of the main components of the Earth's climate system. Its interaction with the atmosphere involves energy fluxes and water and carbon exchanges that are crucial for weather forecasting and climate studies [17–19]. Despite their importance, the land-atmosphere exchanges in land surface models present considerable biases, especially during extreme weather events [20,21]. In the case of simulating latent and sensible heat fluxes, physics-based land surface models were outperformed by simplistic empirical models [22,23].

Climate reanalysis combine model and observations using state-of-the art models and data assimilation techniques. The European Centre for Medium-Range Weather Forecasts (ECMWF) has developed several atmospheric and ocean reanalyses, with the two most recent atmospheric reanalyses being ERA-Interim [24] and ERA5 [25]. The ECMWF reanalyses are generated by the Integrated Forecasting System (IFS), a global data assimilation and forecasting system developed by ECMWF for weather forecasts. The Hydrology Tiled ECMWF Scheme of Surface Exchanges over Land (HTESSEL) [26,27] is the land-surface component of the IFS.

Preliminary results conducted by ECMWF showed ERA5's overall improvement in comparison to ERA-Interim in simulating several different variables. Albergel et al. [28] compared ERA5 and ERA-Interim atmospheric forcing in land surface model simulations, ultimately showing that ERA5 provides an improved product over ERA-Interim. Besides being recent, ERA5 will serve as the official replacement of ERA-Interim, hence it is imperative to evaluate its ability in simulating an ECV such as the LST.

The evaluation of simulated LST using remote sensing LST products has been the subject of analysis in several studies [7,8,29]. Trigo et al. [7] found an underestimation of daytime LST over most of Africa and Europe (especially over semi-arid regions) and a slight LST overestimation during nighttime in the ECMWF model when compared to LSA-SAF's LST. With the same datasets, focusing on Europe but extending the temporal range, Orth et al. [8] also found an underestimation of the LST daily range (especially in the Iberian Peninsula, order of 10 °C). Zhou et al. [29] examined several reanalysis products (including ERA-Interim) over China with the reference-LST measured by in-situ stations. Despite the different regions and analysis performed, these studies suggest that most of the reanalysis underestimate LST, especially during Summer and in arid regions. LST can also be used to guide model development. Trigo et al. [7] presented a revision for different surface parameters (Leaf Area Index (LAI), roughness length for momentum and for heat) and assessed its impact in the simulation of LST. The revised roughness lengths had a positive impact on the daytime LST while the revised LAI had a minor yet positive effect. Orth et al. [8] showed that the LST performance is highly

sensitive to three surface parameters: the minimum stomatal resistance, the skin conductivity, and the soil moisture stress function. Moreover, LST is pertinent in data assimilation. For example, Ghent et al. [30] showed that the assimilation of satellite-LST positively impacts the simulation of LST, soil moisture and the latent and sensible heat fluxes.

The main goals of this study are to (i) evaluate the land surface temperature in the ECMWF ERA-Interim and ERA5 reanalyses over Iberian Peninsula using the LSA-SAF satellite product and to (ii) understand the main drivers of the LST errors in the reanalysis. The study is focused on the summer period (June-August). Simulations with the HTESSEL model in offline mode (uncoupled) were carried out over the Iberian Peninsula and compared with the reanalysis data. Several sensitivity simulations were performed in a confined domain centered in Southern Portugal to investigate potential sources of the LST errors. Our hypothesis is that certain model parameters (e.g., the prescribed vegetation cover) are crucial for the simulation of LST. The following section presents the data and methods, followed by the results and discussion. The overall conclusions of the study are presented in the last section.

## 2. Material and Methods

### 2.1. Models and Datasets

#### 2.1.1. ECMWF Land Surface Model

HTESSEL is the land surface model of ECMWF IFS. It represents the surface skin layer, a shallow layer with zero heat capacity that separates the subsoil from the atmosphere and intercepts and emits radiation. Each grid point of this layer can be divided into different tiles that represent different types of land cover (bare ground, low and high vegetation, intercepted water (on the canopy), and shaded and exposed snow). Only the dominant type of low/high vegetation at each grid point is considered by the model. This information is then used to generate spatial fields of various parameters used in different parameterizations, which are assumed to be dependent only on vegetation type (Table 1). A detailed description of the model assumptions and parameterization can be found on the model documentation [31]. In the following, a more detailed description of the processes directly linked with the simulation of LST in HTESSEL is presented.

The vegetation cover and types are provided to HTESSEL as input static 2-dimensional fields of low vegetation grid fraction (CVL), high vegetation grid fraction (CVH), dominant type of low vegetation (TVL) and dominant type of high vegetation (TVH). Neglecting interception and snow, the low vegetation tile fraction ($C_{low}$), high vegetation tile fraction ($C_{high}$), bare ground tile fraction ($C_{bare}$) and the total vegetation cover of a grid cell (TVC) are given by:

$$
\begin{aligned}
C_{low} &= CVL \times cveg(TVL) \\
C_{high} &= CVHL \times cveg(TVH) \\
C_{bare} &= 1 - C_{low} - C_{high} \\
TVC &= C_{low} + C_{high}
\end{aligned}
\tag{1}
$$

where cveg is the vegetation density (0–1) which is dependent on vegetation type (see Table 1).

These vegetation fields are the same as used in the reanalysis and operational weather forecasts and were derived from the Global Land Cover Characteristics (GLCC) data [32].

The temperature of the skin layer, (LST, also referred as skin temperature), is computed from the surface energy balance equation calculated independently for each tile. The grid-box LST is defined as the weighted average of the LST on each tile fraction. The skin layer is thermally coupled to the four-layer soil below through a conductivity parameter. The skin layer is coupled to the lowest level of the atmosphere using the Monin-Obukhov similarity theory and this coupling is represented by turbulent exchange coefficients (function of atmospheric stability) and the roughness lengths for momentum and heat ($z_{0m}$ and $z_{0h}$, respectively, see Table 1).

**Table 1.** HTESSEL land cover types and associated parameters' values. H/L differentiates low (L) from high (H) vegetation; cveg is the vegetation density (0–1) used in the tile fraction definition; and z0m and z0h are the roughness lengths for momentum and heat, respectively used in the calculations of the turbulent exchange coefficients for momentum, heat and water.

| Index | Land Cover Type | H/L | Cveg | $z_{0m}$ | $z_{0h}$ |
|---|---|---|---|---|---|
| 1 | Crops, mixed farming | L | 0.90 | 0.25 | $0.25 \times 10^{-2}$ |
| 2 | Short grass | L | 0.85 | 0.20 | $0.20 \times 10^{-2}$ |
| 3 | Evergreen needleleaf trees | H | 0.90 | 2.00 | 2.00 |
| 4 | Deciduous needleleaf trees | H | 0.90 | 2.00 | 2.00 |
| 5 | Deciduous broadleaf trees | H | 0.90 | 2.00 | 2.00 |
| 6 | Evergreen broadleaf trees | H | 0.99 | 2.00 | 2.00 |
| 7 | Tall grass | L | 0.70 | 0.47 | $0.47 \times 10^{-2}$ |
| 8 | Desert | - | 0 | 0.013 | $0.013 \times 10^{-2}$ |
| 9 | Tundra | L | 0.50 | 0.034 | $0.034 \times 10^{-2}$ |
| 10 | Irrigated crops | L | 0.90 | 0.50 | $0.50 \times 10^{-2}$ |
| 11 | Semidesert | L | 0.1 | 0.17 | $0.17 \times 10^{-2}$ |
| 12 | Ice caps and glaciers | - | - | $1.3 \times 10^{-3}$ | $1.3 \times 10^{-4}$ |
| 13 | Bogs and marshes | L | 0.6 | 0.83 | $0.83 \times 10^{-2}$ |
| 14 | Inland water | - | - | - | - |
| 15 | Ocean | - | - | - | - |
| 16 | Evergreen shrubs | L | 0.50 | 0.10 | $0.10 \times 10^{-2}$ |
| 17 | Deciduous shrubs | L | 0.50 | 0.25 | $0.25 \times 10^{-2}$ |
| 18 | Mixed forest | H | 0.90 | 2.00 | 2.00 |
| 19 | Interrupted forest | H | 0.90 | 1.1 | 1.1 |
| 20 | Water and land mixtures | L | 0.60 | - | - |

### 2.1.2. ECMWF's Reanalyses

ERA-Interim is an atmospheric reanalysis based on a 2006 version of the IFS (cycle 31r2). Its configuration used a 30-minute time step and a spectral TL255 horizontal resolution (approximately 79 km on a reduced Gaussian grid). The vertical resolution has 60 model layers that reach the top of the atmosphere, located at 0.1 hPa. The surface fields have a three-hourly resolution (eight daily values). ERA5 is the latest ECMWF's atmospheric reanalysis, produced by Copernicus Climate Change Service. It is based on a 2016 version of the IFS (cycle 41r2). The horizontal resolution is about 31 km (TL639). It has 137 vertical layers culminating at 0.01 hPa. The analysis and forecast fields have 24 daily values (hourly output). ERA5 is the official replacement of ERA-Interim, offering a global improvement with several different technical changes [33] and innovations, benefiting from 10 years of model and data assimilation developments by ECMWF. The reanalysis data were extracted from ECMWF data servers in a regular latitude/longitude 0.25° × 0.25° grid. In addition to the LST, the Total Cloud Cover (TCC), that quantifies the percentage of cloud cover in each grid point was also processed.

### 2.1.3. Simulations Setup

HTESSEL is available as an independent library from the atmospheric model (also referred to as "externalized"). HTESSEL's externalization allows it to perform land-surface only (or offline) simulations at a much lower computation cost when compared to full global atmospheric simulations. The offline simulations are driven by near-surface state of air temperature, humidity, wind speed, pressure, solar and thermal downwelling energy, and precipitation [27,28].

In addition to ERA5 and ERA-Interim data, surface only simulations were carried out for a domain centered over the Iberian Peninsula (35°N to 45° N, 10°W to 5°E) with a regular 0.25° × 0.25° resolution. The simulations are initialized in 2002 (from ERA5) to allow the model to spin up (2 years), running for a 14-year interval until the end of 2015 with a 15-minute time step. Initial simulations with a 1-hour time step, which is commonly used, indicated some temporal lag in the LST simulations associated

with the numerical solver. The surface model version used for the offline simulation was CY45R1, which is identical to ERA5 in terms of land surface processes representation.

### 2.1.4. LSA-SAF's Land Surface Temperature

The LST produced by the LSA-SAF is derived from measurements performed by the Spinning Enhanced Visible and InfraRed Imager (SEVIRI) onboard the Meteosat Second Generation (MSG) series of satellites by employing a generalized "split-window" technique [11,12]. This method estimates LST as a linear function of the brightness temperatures at the top of the atmosphere measured by SEVIRI's IR channels centered at 10.8 μm and 12.0 μm. The regression coefficients depend explicitly on the surface emissivity for both channels and implicitly on the total column water vapor and the satellite zenith view angle (SZA).

The LST is available every 15 minutes for all the land pixels of the MSG disk (which comprises SZAs between 0° and 80°), with a resolution of 3 km at the sub-satellite (nadir) point. LST uncertainty is usually between 1–2K, except for regions near the edge of the MSG disk (due to large optical paths associated with high SZAs) or arid areas (where the surface emissivity's uncertainty is generally high, e.g., the Sahara desert) in which the error is larger [13].

The LSA-SAF LST remote sensing product is used to evaluate the quality of both ECMWF reanalysis for the 2004–2015 period. For the comparison between simulated and observed LST to be consistent, we performed an upscaling of the LST data, by computing the median of the whole group of LST pixels within each 0.25° × 0.25° grid cell. The number of original LST data (~5 km of resolution) in each grid cell varied between 30 and 56 pixels. The fraction of valid pixels (each cell and time) was retained to be used as a proxy for cloud cover.

### 2.1.5. Land Cover and Vegetation Datasets

Two different datasets were used in this study: the Copernicus Global Land Service (CGLS) fraction of green vegetation cover (FCover) and the European Space Agency's Climate Change Initiative (ESA-CCI) Land Cover dataset.

The CGLS-FCover represents the fraction of ground covered by green vegetation, which quantifies the spatial extent of the vegetation. The FCover estimates are obtained through a near real-time algorithm that uses top-of-canopy reflectance observations from the SPOT/VEGETATION, and since 2014 from PROBA-V [34]. The product is available globally at 1 km spatial resolution on day 10, 20 and the last day of each month since 1999. Since 2014, a 300 m resolution PROBA-V-only version of the product is also available, but the 1 km (V2) version was considered more practical for the purpose of this study.

The ESA-CCI Land Cover dataset provides globally consistent maps at 300 m spatial resolution on an annual basis from 1992 to 2015. The land cover typology was based on the Land Cover Classification System (LCCS) developed by the United Nations (UN) Food and Agriculture Organization. A total of 22 land cover level 1 classes and 14 level 2 sub-classes (defined using a set of classifiers) constitute the dataset [35]. In this study we used the global map for the year 2010. The data for 1992 was also evaluated, with differences over the study region of the order of 2% when compared with 2010. Both CGLS-FCover and ESA-CCI were aggregated to 0.25° × 0.25° resolution over the Iberian Peninsula domain by mapping each pixel of CGLS-FCover and ESA-CCI to the nearest 0.25° × 0.25° grid-cell. For the CGLS-FCover, the average of all pixels falling in each 0.25° × 0.25° grid-cell was taken. For ESA-CCI, the unique land cover classification at 300m was transformed into land cover classification fractions at each 0.25° × 0.25° grid-cell.

## 2.2. Methods

### 2.2.1. Simulations Evaluation

When comparing the model with the satellite product of LST only clear-sky conditions were considered. Only the data meeting the following clear-sky thresholds was retained in the subsequent analysis:

- The reanalysis' TCC < 0.3;
- The fraction of valid satellite LST original data in each $0.25° \times 0.25°$ grid cell > 0.7.

The two previous thresholds were chosen based on the average percentage of valid data available for the comparison while keeping, at the same time, most of the grid cell cloud-free.

The clear-sky threshold of each reanalysis was also applied to the corresponding HTESSEL offline simulation (driven by that reanalysis) so that all datasets could be compared. The offline simulation forced by ERA-Interim (that has an hourly output) was analyzed using a three-hourly time step in order to match with ERA-Interim's three-hourly output.

The analysis was performed for the period between 2004 and 2015, for which a Climate Data Record (CDR) of reprocessed LST with consistent algorithm and inputs is available from LSA-SAF. The analysis was performed over the Iberian Peninsula (35°N–45°N, 10°W–5°E), ensuring a reasonable variability of land cover types. We only examined the summer months (June-July-August, JJA) due to the overall low percentage of valid data that is available during the rest of the year (e.g., in 2010, before May and after September, the valid data was below 50%). Furthermore, previous studies show that the mean error between reanalyses and observations is higher in the summer [7,29].

An analysis was carried out to separate the domain into different clusters. This clustering is performed to allow summarizing the results by grouping the statistics into regions with similar behavior of the LST. The exercise was applied to different sets of variables (e.g., LST, FCover) in order to identify the most appropriate strategy of clustering the domain but using in all cases the K-Means Clustering Algorithm [36]. The K-Means is a partitional method in which each datapoint belongs to one cluster only, each cluster being comprised of data points with similar characteristics, defined by the input data (i.e., maximizing variability among clusters and minimizing variability within. To ensure that the correct number of clusters was selected, the Elbow Method was applied: the optimal number of clusters (that should be designated into the K-Means algorithm) is that which the addition of an extra cluster would result in a negligible change in inertia (a decrease of less than 10% of its value).

The LST daily maximum (Tmax) and the daily minimum temperature (Tmin) were calculated for the whole domain and for the whole period in the following range of hours (UTC):

- Tmax: 11 h–15 h;
- Tmin: 3 h–7 h.

These ranges were chosen to avoid the identification of Tmax or Tmin in a time period which is not expected (e.g., on cloudy daytime and clear-sky night-time would identify Tmax during the latter). The time range were selected based on the assumption that the maximum temperature will occur shortly after the peak of incoming solar radiation, while the minimum temperature will precede the sunrise. This procedure was applied to both satellite and simulations. The time of occurrence of the Tmax or Tmin is allowed to vary (in the prescribed window) between satellite and simulations, which neglects temporal shift differences.

A set of metrics was chosen to assess the quality of the four different products analyzed in this work: (i) Mean Error (or Bias): computed as the difference between the reanalysis (or model) and the satellite data (model-observations); (ii) the Standard Deviation of the Error (SDE); (iii) the Temporal Correlation and the (iv) Root Mean Squared Error (RMSE)

To conclude the first part of this study, we assessed the relationship between the error in the simulation of LST and the difference between HTESSEL TVC and CGLS-FCover. HTESSEL TVC

is computed from Equation (1), and the CGLS FCover was averaged for the 1999–2018 period in each grid-cell.

## 2.2.2. Sensitivity Simulations

In the second part of the study, we assess potential sources of LST errors associated with the model's representation of vegetation. The domain comprised four grid points in Southern Portugal, near Évora city (38.25°N–38.75°N, 8.25°W–7.75°W) with the same 0.25° × 0.25° resolution as the original domain (see Figure S1 in the supplementary material). The chosen area is representative of the main features and errors explored. These simulations were initialized in January 2009 extending until September 2010, and only the 2010 summer is analyzed. The simulations and analysis focus only on the 2010 summer (that had average conditions in the 2004–2015 period) to reduce the computational cost and data handling.

Several experiments were carried out to investigate the role of vegetation in LST (see Table 2). Three idealized experiments were designed: (1) Bare soil (bare), where CVL and CVH were assigned to zero (in other words, the domain is assumed to be a desert); (2) low vegetation (lveg), where CVL original value was kept and CVH was changed to zero; and (3) high vegetation (hveg), where CVL was altered to zero while CVH kept its original value. These simplified experiments were followed by two other simulations, this time using the CGLS-FCover: (1) nlveg, where CVH was changed to zero and CVL was given the value of the CGLS-FCover, and (2) nhveg, where the CGLS-FCover was attributed to CVH while CVL was altered to zero. In these simulations cveg was assumed to be 1 to guarantee that TVC in the simulations were identical to the CGLS-FCover. The CGLS-FCover considered was the mean for the year 2010.

**Table 2.** Simulations setup configuration for each experiment denoting the used low vegetation grid fraction (CVL), the high vegetation grid fraction (CVH), the vegetation density (cveg) and the total vegetation cover (TVC).

| Experiment | CVL | CVH | cveg | TVC |
|:---:|:---:|:---:|:---:|:---:|
| CTR,9L (SEI [1], SE5 [1]) | IFS [2] | IFS [2] | | |
| bare | 0 | 0 | Table 1 | Equation (1) |
| lveg | IFS [2] | 0 | | |
| hveg | 0 | IFS [2] | | |
| nlveg | CGLS-FCover | 0 | 1 | CGLS-FCover |
| nhveg | 0 | CGLS-FCover | 1 | |
| revised | ESA-CCI [3] | ESA-CCI [3] | Table 1 | Equation (1) |

[1] Simulation for Iberia driven by ERA5 (SE5) and ERA-Interim (SEI). [2] Using IFS vegetation data as in ERA5. [3] Using vegetation cover derived from ESA-CCI land cover.

Following these idealized experiments, we perform a more realistic experiment by replacing the vegetation cover (CVH, CVL) with new fields derived from ESA-CCI (revised). The ESA-CCI Land Cover dataset was converted to Plant Functional Types (PFTs) using the 'cross-walking' table suggested by Poulter et al. [35]. However, the HTESSEL vegetation types (see Table 1) do not have a one to one relation with the cross-walking table used. Instead of performing an ad hoc matching between the cross-walking table PFTs and HTESSEL vegetation types, we kept the model vegetation types (TVL and TVH), and only changed the model CVL and CVH to the values obtained with the ESA-CCI dataset. A more detailed study is required to change the cross-walking table to match the HTESSEL vegetation type. To derive CVL and CVH from the cross-walking table PFTs we aggregated the individual PFTs fraction on each cell considering for CVL: "Shrubs Broadleaf Evergreen", "Shrubs Broadleaf Deciduous", "Shrub Needleleaf Evergreen", Shrubs Needleleaf Deciduous", "Natural Grass",

"Managed Grass" and for CVH: "Tree Broadleaf Evergreen", "Tree Broadleaf Deciduous", "Tree Needleleaf Evergreen", "Tree Needleleaf Deciduous".

Additionally, we tested a different vertical discretization of the soil layers from the original four (7, 21, 72 and 189 cm) to nine (1, 2, 4, 8, 10, 25, 50, 100 and 100 cm layers) (9L), and assessed its impact in the control and in the revised simulations. The original top layer of 7 cm is arranged in 3 layers in the test with 1, 2 and 4 cm which should ensure numerical accuracy of the soil heat diffusion near the surface. This increased vertical resolution at the soil-atmosphere interface was shown to be beneficial when comparing model soil moisture with satellite estimates [37].

Finally, we performed a more detailed sensitivity study on the vegetation density cveg parameter to explore its influence on the LST errors. This analysis was performed with the original and the revised vegetation cover. We perturbed cveg associated with the TVL and TVH within the range 0.1–1, forming 100 perturbations composed of cveg pairs. The pairs were determined with a quasi-random sampling method, the Sobol sequences [38,39], which are designed to efficiently generate samples of the multiple parameters that cover the entire parameter space while avoiding the introduction of correlations between the perturbations of the different parameters. Unlike in random sampling, the sample values in the Sobol approach are selected based on the previously generated values to prevent the occurrence of clusters or empty spaces in the domain.

## 3. Results

### 3.1. Evaluation

The percentage of valid data in summer for the period 2004–2015 (after applying the clear-sky thresholds to both model and satellite LST) is represented in Figure 1. Most of the Iberian Peninsula has over 50% of available data in the 12-year period. Despite this high coverage, there is a clear North-South gradient with southern areas with more than 80% of valid data while the North Iberian Coast and Pyrenees with coverages of only 20%. In other seasons the coverage of valid data is lower (not shown), particularly during winter, which further motivated our decision to focus the evaluation to summertime. If more restrictive thresholds had been chosen, that would have resulted in a reduction of the valid data, but the associated cloud cover values would have remained similar.

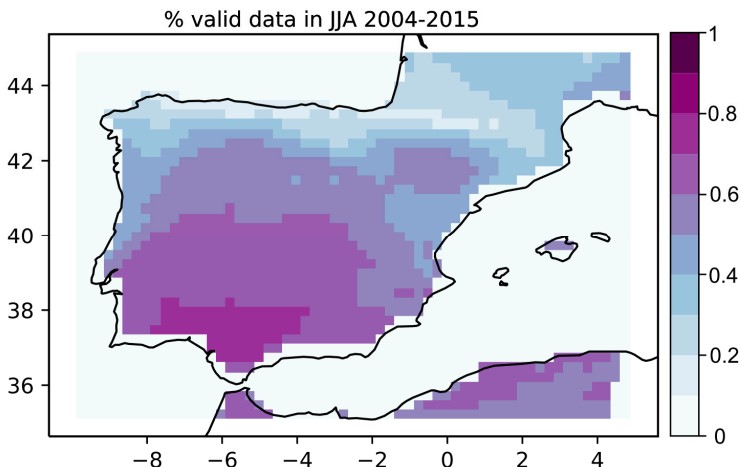

**Figure 1.** Percentage of valid data during summer (June–August) between 2004 and 2015.

The study area was grouped into six different clusters using the K-Means clustering algorithm (Figure 2). The input data of the K-means were the maximum and minimum LST of the mean diurnal cycle in each pixel in the summer months for the period 2004–2015. The six clusters represent regions with different LST diurnal cycles (Figure 2), although some of the regions show similar diurnal cycles in both reanalyses. Clusters 0 (Northern Iberia) 2 (Central Iberian Plateau) and 4, 5 (Southern

semi-arid Iberia) clearly identify different LST diurnal patterns associated with underlying land cover or topography. Clusters 1 and 3 do not clearly identify any land cover or topographical features with a mixture of coastal and inland areas.

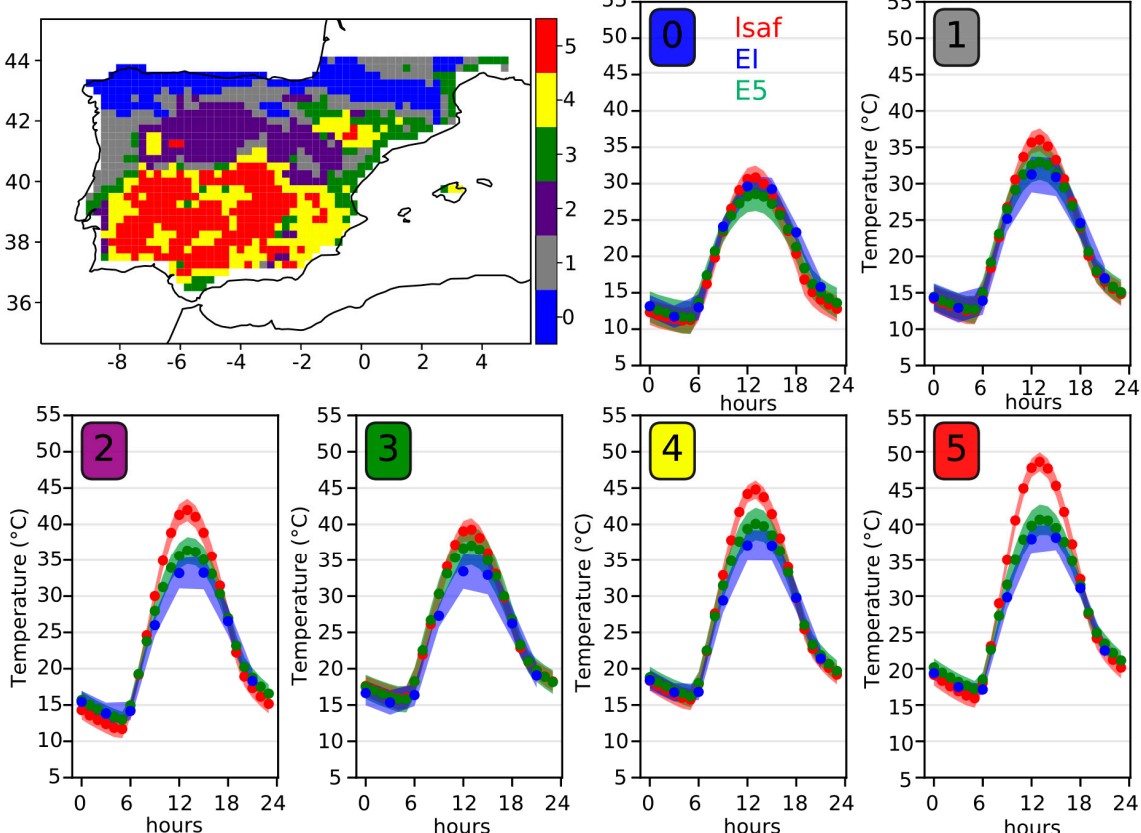

**Figure 2.** Clusters determined by the K-Means Algorithm with Land Surface Temperature (LST) as input (**top left**) and the 2004–2015 mean diurnal cycle (average in dots and Standard Deviation shaded) in each cluster for the satellite-LST (red), ERA5 (green) and ERA-Interim (blue). The number on the top left of each panel identifies the cluster number shown on the top left panel: (**0**) blue, (**1**) grey, (**2**) purple (**3**) green, (**4**) yellow, (**5**) red.

Figure 3 shows the grid-point distributions of the Tmax and Tmin mean error, standard deviation of the error, root mean square error and temporal correlation of the reanalysis and surface simulation. The Tmax bias is consistently negative, with ERA-Interim showing slightly larger bias than ERA5 (Figure 3a). Both surface experiments have similar biases to ERA5, suggesting that the updates in the HTESSEL model (from ERA-Interim to ERA5) had a positive impact on LST, even if its atmospheric forcing is of lower quality (like in the case of the simulation forced by ERA-Interim, SEI). The Tmin biases are generally lower than in Tmax (Figure 3e). As well, for Tmin, the simulation forced by ERA5 (SE5) is systematically colder than ERA5 (about 1 K), while ERA-Interim and SEI are closer and with slightly smaller errors than ERA5.

The Tmax standard deviation of the error (SDE) is around 2–3K, with higher values in ERA-Interim and SEI (Figure 3b). This can explain the better quality of the meteorology dynamics in ERA5 also present in SE5. The Tmin SDE (Figure 3f) is generally smaller than Tmax (with values between 1–2K). ERA5 and both surface simulations have lower errors than ERA-Interim. The RMSE in Tmin (Figure 3g) is much lower than Tmax (median value within 1K and 2K). The distribution of the errors time series (model-observations) are not gaussian, with a positive skewness. However, further evaluation of the errors distribution is not necessary, considering the large systematic (mean) differences found.

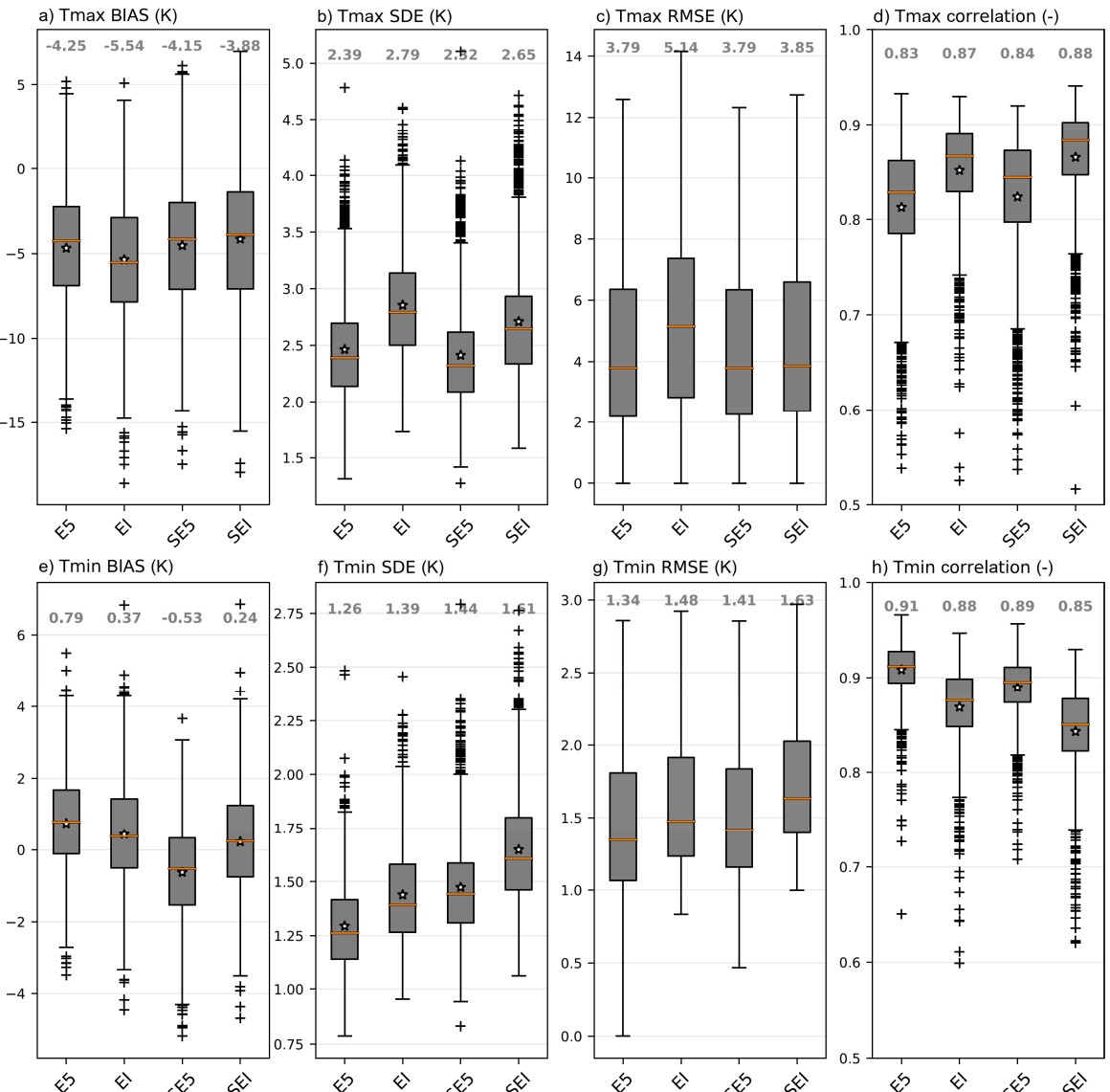

**Figure 3.** Distributions of LST Tmax (Tmin) mean error (**a**,**e**), standard deviation of the error (**b**,**f**), root mean square error (**c**,**g**) and temporal correlation (**d**,**h**) in: ERA5 (E5), ERA-Interim (EI) and the simulations forced with E5 (SE5) and EI (SEI). The red line indicates the median, the filled box represents the interquartile range (25th to 75th percentiles), the whiskers indicate the 10th and 90th percentiles, the white stars the mean, and the cross markers represent outliers. The number above each boxplot is the median of that boxplot. Note the different vertical scales between the Tmax (top) and Tmin (bottom) panels.

The temporal correlation shows very similar values among the four products, for both Tmax and Tmin (Figure 3d for Tmax and Figure 3h for Tmin). Still, it is interesting to notice that, for this metric, ERA-Interim and SEI have consistently better correlations than ERA5 and SE5 in Tmax, while for Tmin ERA5 outperforms ERA-Interim. The coarser (and therefore smoother) temporal and spatial resolution of ERA-Interim might explain these results, but it remains unclear why this is visible only for Tmax and not Tmin.

In Figure 4 the Tmax RMSE are represented for the whole domain (and in Figure 5 the Tmin RMSE). The zones with higher Tmax RMSE are similar among the four products, comprising the South-West Iberian region and the Northern Iberian Plateau, where the RMSE reaches values above 8

or even 10 K in a vast number of grid points. The Tmin RMSE is overall higher in mountainous regions although the zones with higher RMSE differ in each product.

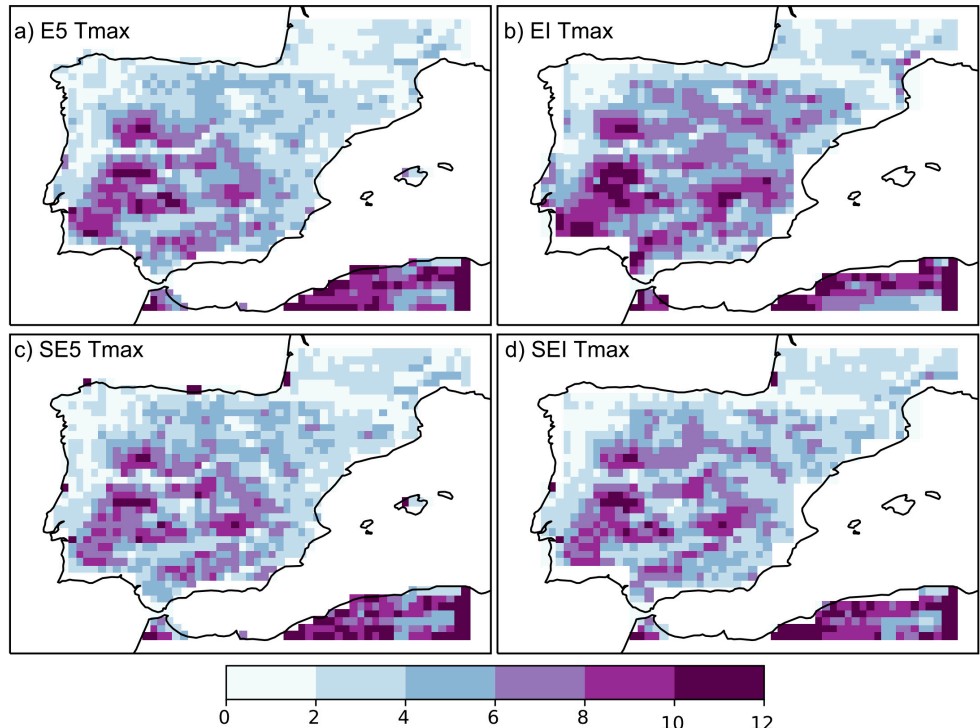

**Figure 4.** Tmax Root Mean Square Error (K) in: ERA5 (**a**), ERA-Interim (**b**) and the simulations forced with ERA5 (**c**) and ERA-Interim (**d**).

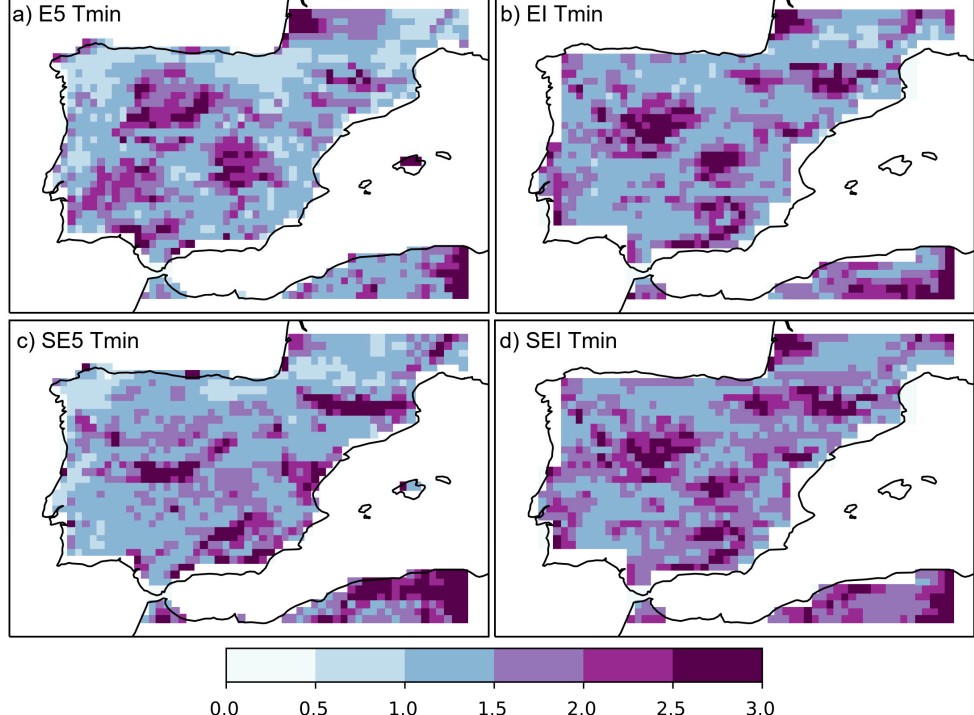

**Figure 5.** As Figure 4 but for the Tmin Root Mean Square Error (K) in: ERA5 (**a**), ERA-Interim (**b**) and the simulations forced with ERA5 (**c**) and ERA-Interim (**d**). Note the different range in the color bars with Figure 4.

Our results are consistent with previous studies [7,8,29], showing a strong daytime underestimation and a weak night-time overestimation of LST in the summer, especially in semi-arid regions. However, our detailed analysis highlights that the daytime errors are not spatially consistent, with some areas in central south Iberia showing much larger errors. Figure 6 displays individual gridpoint mean temperature errors against the respective differences between ERA5 and CGLS-FCover, revealing a negative correlation (−0.45) between the two (Figure S3 shows the CGLS-FCover and ERA5 TVC). These results suggest that the large systematic underestimation of daytime LST can be partially attributed to an overestimation of total vegetation cover in ERA5. This overestimation of vegetation will be reflected in a higher coupling between the skin layer and the atmosphere via turbulent exchanges (higher roughness). This stronger coupling limits the model ability to represent very high daytime LST. In the following section, several sensitivity simulations are examined to further investigate the role of vegetation cover in these large Tmax errors.

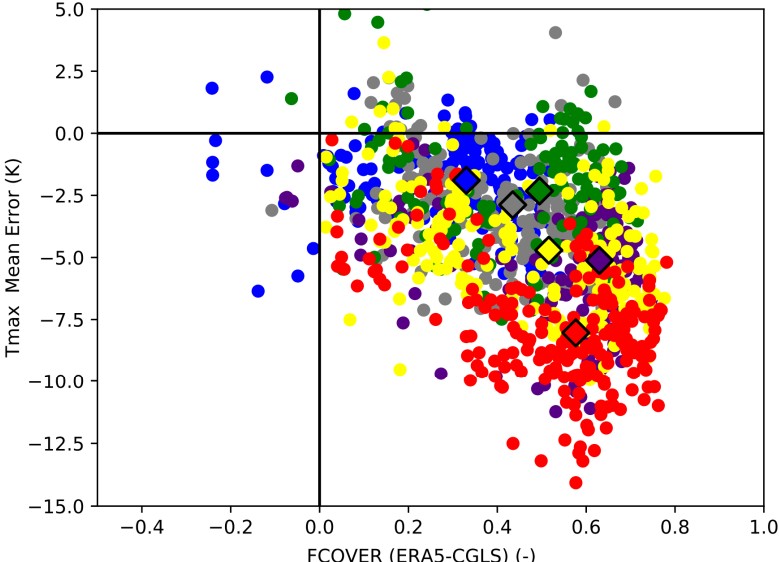

**Figure 6.** Tmax Mean Error (K) in ERA5 as function of the difference between CGLS-FCover and ERA5 Total Vegetation Cover (TVC). The colors represent the clusters in Figure 2. The diamond marker represents the median value of each cluster.

*3.2. Sensitivity Experiments*

3.2.1. Vegetation Cover

The original and revised (derived from ESA-CCI) CVL and CVH in the Southern Portugal domain (see Figure S1) is presented in Table 3, and we determined the associated TVC using Equation (1). In all points, the dominant high vegetation type is 'Interrupted Forest' while the dominant low vegetation type is 'Evergreen Shrubs', with most of the grid point covered by high vegetation (CVH > 0.80 except for the NW point). The percentage of valid data (cloud free) in the domain in the summer of 2010 varied between 0.70 and 0.73, which is similar to the 2004–2015 JJA mean in Figure 1.

The model LST is a weighted average of the different tiles. To further investigate each tile behavior, in Figure 7 we compare the mean diurnal cycle of LST given by each of the models active tiles (low vegetation, high vegetation and bare ground) as well as the underlying soil temperature and overlying forcing air temperature. The LST diurnal cycle in the summer of 2010 of the control simulation presents strong cold biases during the day and weak warm biases at night (Figure 7). The NE and SW points have the warmest satellite-LST diurnal cycles while the NW and SE points have the warmest 'control' LST diurnal cycles. As a result, the NW point (where the HTESSEL-TVC is much lower (0.66) with respect to the remaining points) shows the best approximation of the model simulation to the satellite observations (with a negative bias of 4.5 K at 12 UTC). It is also the only point where the bare ground

LST (the warmest LST) comes very close to the satellite-LST. The high vegetation tile LST is very similar to the control LST in all the points while the low vegetation LST has the lowest values, which is attributed to CVH and CVL having values close to one and zero, respectively. The control LST diurnal cycle presents a slight phase difference in relation to the satellite-LST during the day, taking longer to warm in the morning and to cool down in the afternoon.

**Table 3.** Revised and original (between brackets) vegetation parameters in in the four points domain in Southern Portugal. CVL and CVH identify the low and high vegetation grid fraction, IFS TVC is the derived Total Vegetation cover (using the cveg in Table 1 and Equation (1)) and the CGLS FCover the fraction of green vegetation from the Copernicus Global Land Service.

| Point | CVL | CVH | IFS TVC | CGLS FCover |
|-------|-----|-----|---------|-------------|
| NW | 0.87 (0.40) | 0.13 (0.51) | 0.55 (0.66) | 0.42 |
| NE | 0.89 (0.07) | 0.08 (0.93) | 0.52 (0.87) | 0.41 |
| SW | 0.89 (0.01) | 0.10 (0.99) | 0.54 (0.89) | 0.47 |
| SE | 0.85 (0.17) | 0.12 (0.81) | 0.53 (0.81) | 0.40 |

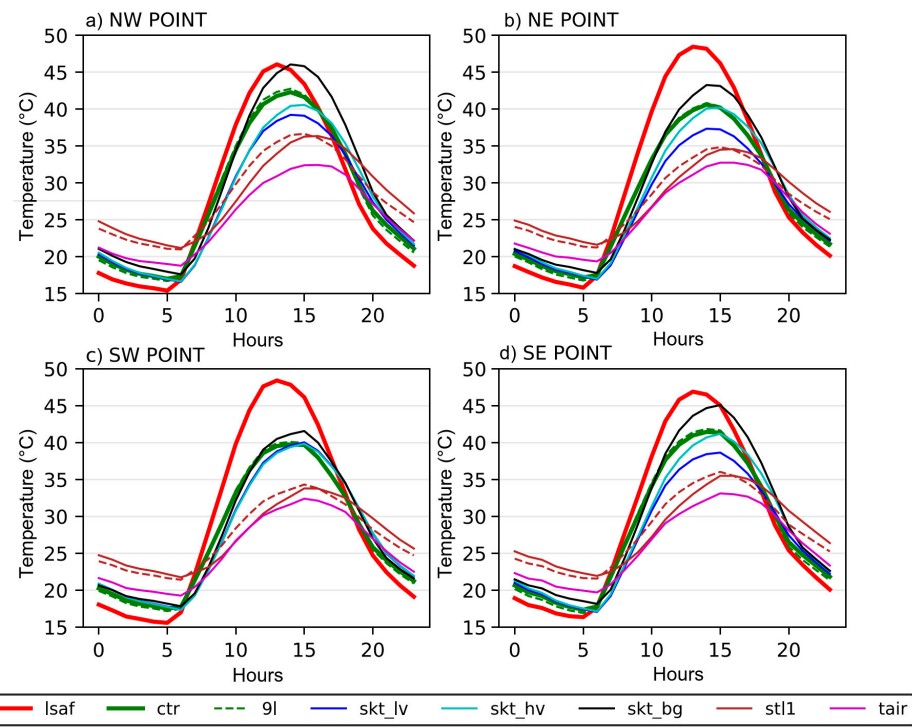

**Figure 7.** Mean diurnal cycle of temperature (2010 Summer) in the 4 points of the Southern Portugal domain (**a**) North West, (**b**) North East, (**c**) South West (**d**) South East (see Figure S1) comparing the satellite LST (red), and the LST in the control simulation (green), and control with 9 soil layers (dashed green). The control tile temperatures of low vegetation (blue), high vegetation (cyan) and bare ground (black) are also represented along with the air temperature at 10 m height used to drive the model (magenta) and simulated first soil layer temperature (stl1, brown).

Air temperature is very similar in all the points, with lower values than the LST during the day and slightly higher values at night (Figure 7). The first soil layer temperature is highest in the NW point (where the control LST is also highest) and follows a similar diurnal cycle to the air temperature, but with higher temperatures overall. The 9 soil layers experiment (9L) did not changed significantly the control LST (dashed green line), leading to a bias decrease of ~0.5 K, but it shows a reasonable variation in the first soil layer temperature (dashed brown line). In the 9L simulation, the first soil

layer only has 1 cm while the control has 7 cm, resulting in a reduction of the thermal inertia in 9L with a faster warming (dashed versus solid brown lines in Figure 7).

The results of the sensitivity simulations (see Table 2) are presented in the supplementary information in Figure S4. The two experiments with CGLS-FCover used as TVC in the model (nlveg and nhveg) present a diurnal cycle closer to the satellite observations, in particular the nlveg simulation, suggesting that TVC should be reduced. The sensitivity simulations with the original model parameters (bare, lveg and hveg) provide a similar conclusion as well: the bare and lveg (the hveg and control) simulations are very similar to each other due to CVL (CVH) being close to zero (one) in all the points except for the NW one, with all experiments remaining distant to the satellite-LST.

The revised vegetation from ESA-CCI (Figure 8) has a positive impact on LST. During daytime, the LST becomes very close to the satellite-LST, with a negative bias below 2K at 12 UTC in all the points, except the NE one. This latter grid box contains the largest urban area (the city of Evora, see Figure S1) amongst the four considered here, which very likely explains the high satellite Tmax value and the largest deviations between the revised simulation and satellite temperatures. The bias becomes positive in the afternoon (but with an absolute value lower than in the control simulation), as the simulation continues to show a phase difference in relation to the satellite-LST. At night, the impact is negligible. Parallel to the control simulation, the soil discretization scheme produced a slightly positive effect in the revised simulation, reducing the bias at night by 0.5–1 K and at midday by ~1K (Figure 8). These results show that vegetation cover dominates over the soil vertical discretization in terms of addressing the large LST biases.

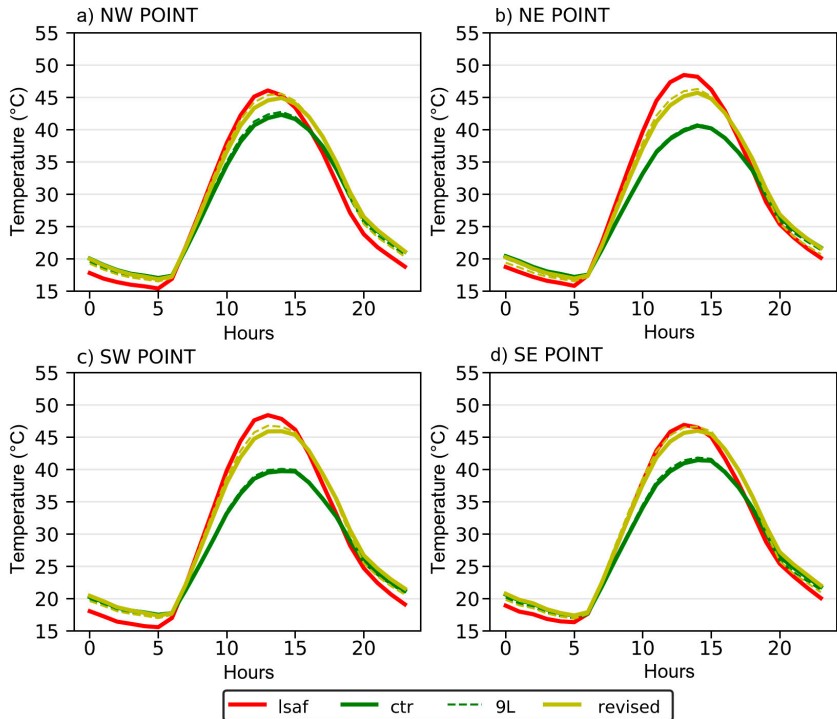

**Figure 8.** Mean diurnal cycle of temperature (2010 Summer) in the 4 points of the Southern Portugal domain (**a**) North West, (**b**) North East, (**c**) South West (**d**) South East (see Figure S1) comparing the satellite LST (red), and the LST in the control simulation (green), and control with 9 soil layers (dashed green) with the simulation using ESA-CCI vegetation cover (revised in yellow, and with the 9 soil layers dashed yellow).

To further understand the changes in LST arising from the vegetation cover changes, we focus on the diurnal cycle of the surface energy balance. The components of the surface energy balance of the control simulation are represented in Figure S5. Not surprisingly, the surface receives mainly

shortwave radiation (SWnet > 0) and emits mainly longwave radiation (LWnet < 0). The sensible (Qh) and latent (Qle) heat fluxes are mostly negative (the heat and moisture transports happen from the ground to the atmosphere), with Qh being slightly positive at night (the heat transport is towards the ground, see also air temperature and soil temperature mean diurnal cycles in Figure 7). The net flux (NET) is positive during the day (the surface warms—energy sink of the atmosphere) and negative at night (the surface cools down—energy source to the atmosphere).

A comparison between the energy components of the control and the revised simulations is available in Figure 9. Longwave upward radiation (LWup) becomes more negative (the surface emits more radiation) in the revised simulation, because LWup follows the Stefan-Boltzmann's Law and LST is higher in the revised simulation. The opposite happens to Qh, as it becomes less negative (less transport of heat from the surface). Since the absolute value of Qh decreases, despite the increased gradient between the air and the skin, it means that the changes in the turbulent transfer coefficients $C_H$ (in this case, a decrease) impacts Qh more than the increase in LST. $C_H$ depends on $z_{0m}$ and $z_{0h}$ and these two parameters have lower values in "Evergreen shrubs" than in "Interrupted forest" (Table 1). The TVL is dominant over TVH in the revised vegetation, which explains the decrease in value of $C_H$ when compared to the original vegetation. The Qle from the surface rises in the revised simulation. The LST increase results in an exponential increase of the vapor pressure at saturation (computed using the model LST) due to the Clausius-Clapeyron relation, which in turn leads to an increase of Qle. The net flux diurnal cycle amplitude increases with increased storage during the day and latter release at nighttime.

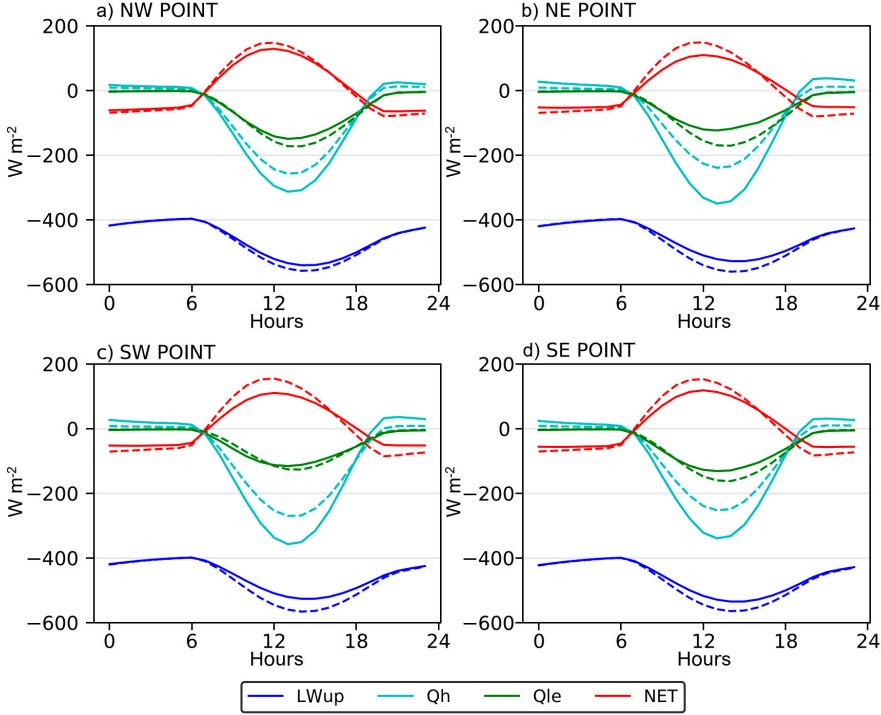

**Figure 9.** Components of the surface energy balance at the surface in the control (solid lines) and revised (dashed lines) simulations. Longwave upward radiation (LWup, blue), sensible heat flux (Qh, cyan), latent heat flux (Qle, green) and the surface net flux (NET = SWnet + LWnet + Qh + Qle, solid red). The fluxes sign conventions indicate fluxes to the surface as positive and fluxes leaving the surface as negative. The downward and net solar radiation (SWdown, SWnet) and longwave downward radiation (LWdown) are identical in both simulations and are represented in Figure S7. Each panel represents one of the 4 points of the Southern Portugal domain (**a**) North West, (**b**) North East, (**c**) South West (**d**) South East (see Figure S1)

### 3.2.2. Vegetation Type

In the previous analysis, only vegetation cover was considered, keeping the vegetation types as in the original IFS. In the selected 4 grid-points the low vegetation type in HTESSEL is "Evergreen Shrubs" while in ESA-CCI's PFTs is "Managed Grass" (see Figure S6 for a comparison of the vegetation types between IFS and ESA-CCI). Since IFS does not have a one to one relation with ESA-CCI's cross-walking table PFTs, it was decided to keep the original vegetation types. With the vegetation from ESA-CCI the dominant cover is low vegetation. To further investigate the role of vegetation type, several simulations were performed changing the default low vegetation type ("Evergreen Shrubs") to all other low vegetation types available in IFS. These changes did not improve the diurnal cycle of LST, with the original type performing better (see Figure S7).

In Figure 10a, the perturbations with lower Tmax errors have very low values in the high vegetation cveg and the perturbations with higher cveg show a consistent underestimation of Tmax, which indicates once more that the original CVH in the model was too high. The low vegetation cveg can take nearly any value since its influence in the LST is reduced due to the low CVL in the original model vegetation. The sensitivity of the cveg parameter for the original CVL and CVH shows no correlation between the TVC and Tmax mean error (Figure 10c), as the same TVC leads to different values of Tmax errors.

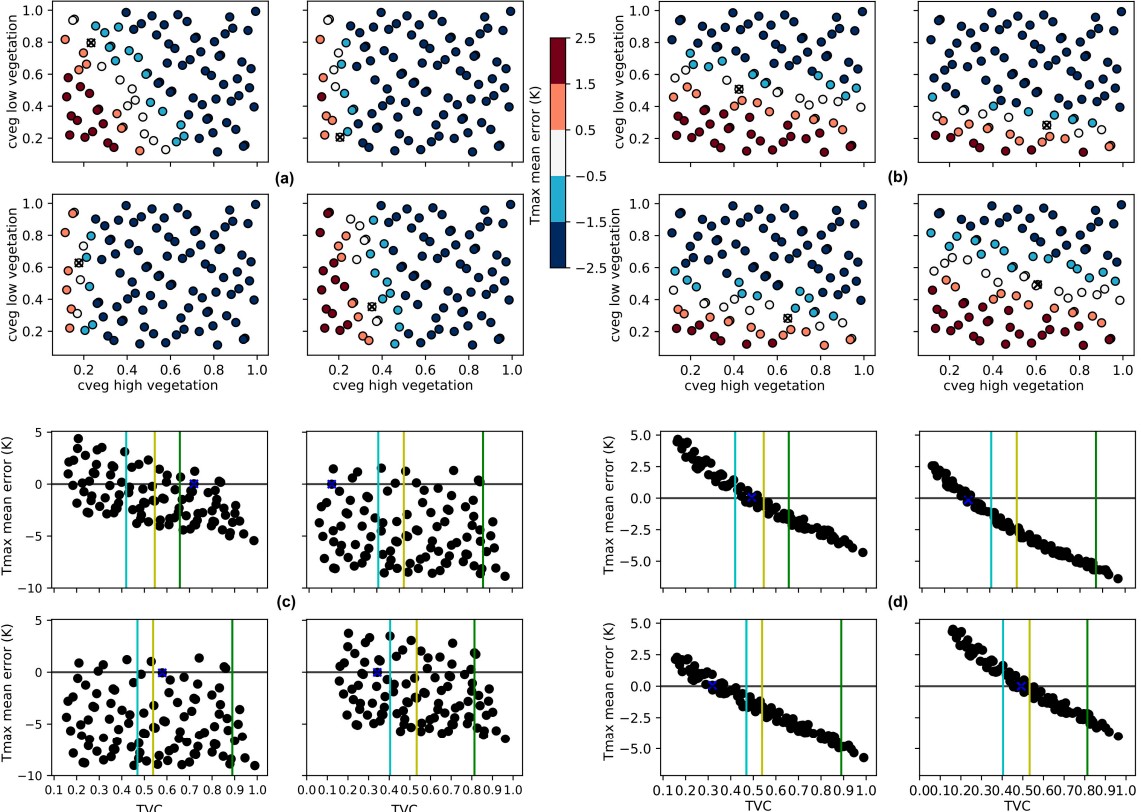

**Figure 10.** Vegetation density (cveg) 100 perturbation pairs (high vegetation x-axis, low vegetation y-axis) and associated Tmax mean error (K) of the (**a**) original and (**b**) revised vegetation cover using ESA-CCI. The cross marks the best pair (with Tmax bias closer to zero). Scatterplots of the Tmax mean error (y-axis; K) as function of the total vegetation cover (TVC) (x-axis) of the 100 perturbations: with the (**c**) original and (**d**) revised vegetation cover using CCI. The blue cross marks the best perturbation (with Tmax Mean Error closer to zero). The vertical lines indicate the TVC in control (green), reviser (yellow) and CGLS-Fcover (cyan). The 4 sub-panels in each panel denote the 4 grid-points in the domain (top left) North West, (top right) North East, (bottom left) South West (bottom right) South East (see Figure S1), sharing the same vertical and horizontal axes.

The sensitivity study applied to the cveg parameter for the revised CVL and CVH presents a considerable correlation between the TVC and Tmax mean error (Figure 10d). In all 4 points the TVC with lower errors is very close to the CGLS-FCover. In Figure 10b, the perturbations with lower errors have, in general, values of low vegetation cveg between 0.4 and 0.6 (except for the NE point, which was the point that contained Évora City), which are similar to the original TVL's cveg of 0.5. The high vegetation cveg can practically take any value because, like CVL in the original vegetation, the CVH in the revised vegetation is small.

## 4. Discussion

The K-Means Algorithm was tested with different input data, namely the CGLS-FCover and both LST and CGLS-FCover data (see supplementary material Figure S2). For the LST+FCover clusters, the LST data was normalized by simply dividing the data by the maximum temperature in the dataset (since there were no sub-zero temperatures in the dataset). In general, the results were similar but the FCover 'inland' clusters (clusters 3 and 4 in Figure S2a) differed from the LST 'inland' clusters (clusters 2, 3 and 4 in Figure 2): There was no separation between the areas to the north and to the south of the Central System in the FCover clusters. While the vegetation cover is similar in both areas (Figure S3), the area to the north of the Central System, the Iberian Plateau, is located at a higher altitude than the area to the south. This cluster analysis was only used to group the mean diurnal cycles in Figure 2. Therefore, it is not expected that a different clusters selection (as for example in Figure S2) would change the main interpretation of the results.

Although ERA5 exhibits better results than ERA-Interim overall (and SE5 in relation to SEI), the surface experiments (see Figure 3) show a lower performance in terms of Tmin. The inconsistency in performance between the offline simulations and the original reanalysis for the Tmin mean error and SDE is likely related to the representation of stable conditions in the offline model, which do not reproduce exactly the coupled system. In the coupled model the vertical diffusion in the atmosphere has an internal half-timestep for numerical stability which is not performed in the offline model. This might introduce numerical differences, which are more evident in stable conditions associated with the computation of the turbulent exchange coefficients. Despite these differences, our results show that the offline simulations reproduce very closely the reanalysis and can therefore be used to investigate potential sources of the errors in ERA5 with a much lower computational cost.

The revised vegetation using ESA-CCI shows a stark contrast to the original model vegetation over the selected domain (see Table 3). Overall, the TVC is much lower (~0.53) and the grid points are covered mostly by low vegetation (CVL > 0.85) when using the ESA-CCI dataset. When comparing HTESSEL's revised TVC to CGLS-FCover, even though CGLS is lower in all the grid points they are closer to each other than to the default HTESSEL TVC. These results follow the preliminary findings with the sensitivity experiments. The simulations using ESA-CCI vegetation cover but different vegetation types (see Figure S7) did not improved the diurnal cycle of LST, when compared with the original type ("Evergreen Shrubs"). These results can be partially explained by the cveg used in HTESSEL for "Evergreen Shrubs" (0.5, see Table 1) which is lower than the cveg of the remaining TVLs. This results in an increase of TVC when changing the TVL to another type. The TVH in HTESSEL is "Interrupted Forest" and in ESA-CCI is "Tree Broadleaf Deciduous" (Figure S6). In this case the cveg parameter would be the same, but with changes in the roughness lengths. In addition to the differences in cveg, the momentum and heat roughness lengths also change when changing the type of low vegetation. The tests to the type of low vegetation indicate that the good performance of the revised simulation on these 4 points is also due to the underlying type of vegetation in IFS, which does not match ESA-CCI. These results suggest the potential role of the cveg parameter, associated with each vegetation type, acting directly on the TVC with impacts on the simulated LST. This motivated a more detailed sensitivity analysis to cveg, as explained in Section 2.2. The sensitivity evaluation further highlights the importance of the representation of vegetation in the IFS, showing a reasonable correlation between cveg and the Tmax bias with the revised model vegetation, while no correlation is

discernible when considering the original model vegetation. These results might explain why past studies that identified the LST Tmax bias in ECMWF products [7,8] did not clearly identify vegetation cover as a plausible cause of the errors.

## 5. Conclusions

The main goal of this study was to evaluate the LST of two ECMWF reanalysis (ERA-Interim and ERA5) using satellite-LST over the Iberian Peninsula. We found a general underestimation of daytime LST and slightly overestimation at night-time. In line with previous studies, the results indicate that there is still room for improvement in the simulation of LST in ECMWF products, with ERA5 presenting an overall higher quality product in relation to ERA-Interim. However, previous studies [7,8] did not investigate in detail the role of the underlying vegetation cover in explaining these errors. Our analysis suggests a relation between the large daytime cold bias and vegetation cover. These results motivated a more detailed evaluation of offline simulation with the ECMWF land-surface model HTESSEL. These simulations were driven by ERA5, and despite some differences, reproduced very closely the main errors in ERA5.

Focusing on a small domain in Southern Portugal, several sensitivity simulations were performed to investigate the role of vegetation cover and the vegetation density parameter on the LST errors. The replacement of the low and high vegetation cover by those of ESA-CCI provided an overall reduction of the large Tmax biases. The increased vertical resolution of the soil at the surface, has a positive impact, but much smaller when compared with the vegetation changes. The sensitivity of the vegetation density parameter, that currently depends on the vegetation type, provided further proof for a needed revision of the vegetation in the model, as there is a reasonable correlation between this parameter and the Tmax mean errors with the revised model vegetation (while the same correlation cannot be reproduced with the original model vegetation).

Despite the overall consistency of our results, this study has several limitations. Although the new vegetation cover leads to improved LST results, the vegetation types of the ESA-CCI dataset is not a direct match to the model and experimenting with different vegetation types translates to poorer results, which are due to the vegetation density parameter. The phase difference and the nighttime bias observed in the LST mean diurnal cycle remain after applying the changes in vegetation and soil discretization. We only performed uncoupled simulations to assess the impact of surface parameters in the simulation of LST. It is important to study the effect of these vegetation changes in coupled simulations as well. Additionally, due to the satellite LST relying on IR measurements, the LST is only assessed in clear sky weather conditions and, therefore, conclusions may be somehow limited. Nevertheless, using clear-sky observations allows focusing the analysis on the deficiencies of the representation of surface parameters, as there are less variables to be accounted for in the surface energy balance (such as clouds and precipitation). Furthermore, changes in vegetation cover impact the water budget (which was not assessed) and induce changes in other seasons. The reduced satellite LST availability in the rest of the year and the lack of other observations (e.g., fluxes, soil temperature, soil moisture) limit further diagnostics.

Our results suggest that vegetation cover is the main contributor to the large daytime biases in LST over Iberia, motivating the need to review the treatment of vegetation cover over the Iberian Peninsula (and most likely over other regions, which have similar climate and phenology), namely the fraction of low and high vegetation cover in each grid point. Likewise, the definition of the different types of low and high vegetation in the HTESSEL and the associated vegetation density parameter and roughness lengths for momentum and heat might also need to be revised. However, we also found a clear problem of equifinality between low and high vegetation cover and the vegetation density parameter, which is challenging for parameter optimization. Furthermore, the current assumption of a constant vegetation density might be also a limitation, since it disregards important seasonal changes in vegetation cover seasonality [40].

The uncoupled simulations allowed us to assess the influence of surface parameters in the LST simulation and the surface energy balance components by varying the value of those parameters (since other factors such as the atmospheric variables remained the same in every experiment). Still, it is important to mention that applying these changes in surface parameters in coupled atmosphere simulations might result in a less positive impact in the simulation of LST, due to feedback processes associated with the atmospheric coupling. In particular, the vegetation cover changes will impact the momentum, heat and moisture exchanges via the changes in roughness lengths.

Finally, it is worth mentioning that model assessments, together with potential revision of model parameters, such as those performed here, are only possible due to the availability of high quality (in terms of their accuracy, temporal span and resolution, and of their spatial sampling) satellite retrieved datasets of Essential Climate Variables. Although variables such as Land Cover and Vegetation Cover have long demonstrated their added value for model development activities, this study clearly shows that Land Surface Temperature can also be used to physically constrain land surface models, which are a key component of Earth System Models.

**Supplementary Materials:** The following are available online at http://www.mdpi.com/2072-4292/11/21/2570/s1, Figure S1: Google-Earth view of Iberian Peninsula (top) and location of the 4 grid-points of the Southern Portugal domain (bottom), Figure S2: Clusters determined by the K-Means Algorithm using as input: (a) CGLS-FCover and (b) both LST and CGLS-FCover, Figure S3: ERA5 Total Vegetation Cover (TVC, left) and the mean 1999-2018 CGLS FCover (right), Figure S4: Mean diurnal cycle of temperature (2010 Summer) in the 4 points of the Southern Portugal domain (a) North West, (b) North East, (c) South West (d) South East comparing the satellite LST (red), and the LST in the control simulation (green), with several sensitivity experiments (see Table 2): bare (black), hveg (cyan), lveg (blue), nhveg (dashed cyan) and nlveg (dashed blue), Figure S5: Surface energy balance in the 4 points of the Southern Portugal domain (a) North West, (b) North East, (c) South West (d) South East of the control simulation (W m$^{-2}$): shortwave downward radiation (SWdown, dashed black), shortwave surface net radiation (SWnet solid black), longwave downward radiation (LWdown, dashed blue), longwave net surface radiation (LWnet, solid blue), sensible heat flux (Qh, solid cyan), latent heat flux (Qle, solid gree) and the net flux (NET=SWnet+LWnet+Qh+Qle, solid red). The fluxes sign conventions indicate fluxes to the surface as positive and fluxes leaving the surface as negative, Figure S6: ERA5 (default IFS) type of high vegetation (top left) and type of low vegetation (bottom left) and ESA-CCI derived dominant type of high vegetation (top right) and dominant type of low vegetation (bottom right), Figure S7: Mean diurnal cycle of temperature (2010 Summer) in the 4 points of the Southern Portugal domain (a) North West, (b) North East, (c) South West (d) South East comparing the satellite LST (red), and the LST in the control simulation (green), revised (dark yellow) and several different types of low vegetation: short grass (sgrass, blue), tall grass (tgrass, dashed brown), crops (magenta) and irrigated crops (irrcrops, dashed yellow). and nlveg (dashed blue).

**Author Contributions:** Conceptualization, S.E. and E.D.; formal analysis, F.J.; funding acquisition, E.D.; writing—original draft, F.J.; writing—review and editing, S.E., J.P.A.M., I.F.T., M.N. and E.D.

**Funding:** This research was funded by Fundação para a Ciência e a Tecnologia (FCT) grant number PTDC/CTA-MET/28946/2017 (CONTROL). E. Dutra was funded by FCT research grant IF/00817/2015. F. Johannsen and M. Nogueira were funded by FCT project CONTROL (PTDC/CTA-MET/28946/2017).

**Acknowledgments:** The authors acknowledge two anonymous reviewers for their comments and suggestions.

**Conflicts of Interest:** The authors declare no conflict of interest. The funders had no role in the design of the study; in the collection, analyses, or interpretation of data; in the writing of the manuscript, or in the decision to publish the results.

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
