# Peer review of "Cold Bias of ERA5 Summertime Daily Maximum Land Surface Temperature over Iberian Peninsula"

_remotesensing, doi:10.3390/rs11212570_

Round 1
Reviewer 1 Report
please see the attached file

Reviewer 2 Report
This was a well-written paper, with careful attention to detail. I recommend publication, with some minor points to consider:
Vegetation type has a significant impact on the results. You considered only the dominant vegetation type. Does the heterogeneity of the vegetation play any role in uncertainty? Line 123: CY45R1 is "very close to the model version of ERA5". What are the differences. Are they significant? How are all datasets resampled to the 0.25 x 0.25 deg resolution in section 2? Similarly for the 3-hour temporal resolution. Mean and standard deviation are metrics to measure the difference between reanalysis and model-observations. Is the difference Gaussian? If it isn't, you may need more metrics to better describe the distribution. I don't see the benefit to the K-means clustering, since two of the classes were clearly mixtures. I think it would be better use of space to include only summary statistics for the entire region, and then include the Tmin figures in the main body, rather than the supplemental material, since they feature so heavily in the discussion.
Round 2
Reviewer 1 Report
The authors addressed all of my concerns hence I recommend publication of the manuscript as is.
However, the 11th reference need attention.